# The Origin(s) of LUCA: Computer Simulation of a New Theory [note 1]

**DOI:** 10.3390/life15010075

**Published:** 2025-01-10

**Authors:** Shiping Tang, Ming Gao

**Affiliations:** Center for Complex Decision Analysis, Fudan University, Shanghai 200433, China; gaoming@fudan.edu.cn

**Keywords:** computer simulation, LUCA, Carl Woese, cellular evolution, global inventions

## Abstract

Carl Woese’s thesis of cellular evolution emphasized that the last universal common/cellular ancestor (LUCA) must have evolved by drawing from “global inventions”. Yet, existing theories regarding the origin(s) of LUCA have mostly centered upon scenarios that LUCA had evolved mostly independently. In an earlier paper, we advanced a new theory regarding the origin(s) of LUCA that extends Woese’s original insights. Our theory centers upon the possibility that different vesicles and protocells can merge with and acquire each other as a form of variation, selection, and retention, driven by wet-and-dry cycles and other similar cyclical processes. In this paper, we use computer simulation to show that under a variety of simulated conditions, LUCA can indeed be produced by our proposed processes. We hope that our study can stimulate laboratory testing of some key hypotheses that vesicles’ absorption, acquisition, and merger has indeed been a central force in driving the evolution of LUCA.

## 1. Introduction

The coming of the Last Universal Common (*Cellular*) Ancestor (LUCA) was a major transition in the making of the biotic world [1,2,3,4] (Although LUCA has been conventionally taken to be the “Last Universal *Common* Ancestor”, it is now generally accepted that LUCA must have been a fairly complete cell [3,4]). Recently, Tang advanced a new theory regarding the origin of LUCA [5]. His core thesis is that vesicle absorption, acquisition, and fusion via breaking and repacking, proto-endocytosis, proto-endosymbiosis, and other similar processes, driven by “wet-and-dry” cycles or other similar thermochemical cycles within Darwin’s “warm little pond(s)” [6,7,8,9], has been a central mechanism that propelled the origin(s) of the First Universal *Cellular* Ancestors (FUCAs) and then the evolution of LUCA from FUCAs. Hence, our position critically differs from Prosdocimi and his colleagues [10,11,12], who also used the abbreviation FUCA for “the First Universal *Common* Ancestor”. Prosdocimi et al. insist that FUCA is not a cell but an RNA-dependent peptide synthesis machinery or peptidyl transferase center (PTC). We believe it is more reasonable to assume that FUCAs were already proto-cells (vesicles), with a more or less functional cell membrane [13,14]. Indeed, according to our theory [5], only if FUCA is a proto-cell can LUCA emerge, partly because vesicles are necessary for concentration, crowding, and extensive molecular interaction.

Tang further hypothesized that the evolution of LUCA came in two major stages: the evolution of FUCAs, and then evolution of FUCAs to LUCA [5]. What is common to both stages has been that both involve the same process of absorption, acquisition, and merger by these protocells (or vesicles), and this process had been the key behind Carl Woese’s thesis that LUCA must have evolved by drawing from “global inventions” [15,16,17]. The whole evolutionary process from simple vesicles to FUCAs and then LUCA is summarized in Figure 1 below (reproduced from [5]).

Tang has provided some chemical and biological evidence that supports his hypothesis [5]. Ultimately, however, chemical and biological evidence can only show that the hypothesis is possible but not necessarily viable for the making of LUCA because recreating LUCA from scratch in a lab is highly unlikely.

We therefore turn to the next best thing: to “visualize” the viability of the proposed hypothesis with computer simulation, following similar efforts with different focuses [18,19,20,21]. More specifically, we aim to show that merger and acquisition by vesicles and other similar processes can indeed produce FUCAs, and then inevitably, LUCA.

Before proceeding further, three caveats are in order.

First, we are mostly interested in simulating the hypothesis that FUCAs and LUCA had most likely evolved by drawing from “global innovation” via vesicles’ merger and acquisition [5,15,16,17,22]. As such, we do not simulate the chemical synthesis of polymers from nucleotides or amino acids [23,24], the evolution of autocatalytic cycles, the evolution of an RNA self-replication system [25,26,27], the formation of an RNA–peptide world [28,29,30], the forming of vesicles [18,19,20], the division of vesicles driven by absorption of ingredients from the environment or merging with other vesicles [9], the coupling of an RNA replication system to vesicles [31], or the coupling of cellularity to metabolism [21,32,33], partly because they have been performed earlier, either via simulation or via wet experiments. Also, simulating the whole process from chemical evolution to the coming of LUCA will be a daunting, if not an impossible, task, without necessarily shedding much light on the central mechanism we highlight here (for instance, Yin et al. [31] simulated how an RNA replication system can evolve and then spread, and their “cellular” evolution is really about vesicles engulfing an RNA replication system). Thus, their simulation does not deal with the evolution of LUCA, or even the evolution of FUCAs. Our simulation (based on our theory) subsumes vesicles engulfing ingredients such as amino acids, nucleotides, peptides, and RNAs. Most critically, Yin et al. [31] based their simulation on the “hydrothermal vent” scenario, which many others and we believe to be rather unrealistic [5,6,7,8]). We therefore focus on the late stage of the evolution of LUCA, somewhat similar to Takagi et al. [21] and Nunes Palmeira et al. [32].

Second, we concur with several earlier studies that LUCA has at least three hallmarks: (1) a fully functioning membrane; (2) a fully evolved standard genetic code (SGC); (3) about one hundred proteins and several hundred genes [34,35,36,37,38,39] (most likely, these genes had been linked into a chromosome [40]). More recent estimation has put the number of proteins within LUCA at 2600, comparable to modern prokaryotes [41]. Here, we use the more conservative estimates. In our simulation, we focus on the coming together of one hundred peptides (plus several hundred genes) and the coming of the SGC while assuming that LUCA has a fully functioning membrane.

Third, we do not engage in the ongoing debates on more specific puzzles of the origins of life (e.g., which comes first? RNA world, RNA–peptide world, metabolism, membrane, etc.), mostly because these puzzles cannot possibly be resolved any time soon (for an excellent survey of the debates, see [42]).

## 2. Materials and Methods

Computer simulations are performed with the GAMA simulation platform, version 1.8.1 (https://gama-platform.org/ (accessed on 20 October 2022)), on local servers hosted by our center. All computer codes and experimental data are available online (https://github.com/xymbgm/2023-05-Tang-and-Gao-Simulation-of-from-FUCA-to-LUCA-codes (accessed on 1 October 2024)).

Each of the simulations described below (i.e., whether the two stages of evolution from simple vesicles to FUCAs and then LUCA are simulated separately or together) starts with one single “warm little pond” (see below for details). All simulations follow a similar flow (see Figure 2 for the flowchart), with one “cycle” in a simulation run being one wet phase or one dry phase, alternated. Table 1 summarizes the parameters in the simulation.

**Table 1 life-15-00075-t001:** Parameters in the simulations.

Parameters	Explanation	Value
tickCount	count of cycles within a simulation	1 cycle = 1 wet phase or one dry phase, alternated
currentVolume	units of volume of a pond	dry phase: (50, 80), wet phase: (80, 100); initiation: 100
numPeptide	number of peptide	see Table 2 for specific values
numRNA	number of RNA	see Table 2 for specific values
numAA	number of AAs in a peptide	see Table 2 for specific values
numNBase	number of nucleotides in an RNA	3 × numAA
V1InitNum	initial number of V1	3000
V2InitNum	initial number of V2	2000
totalnum	external supply of V1 and V2 in every cycle	dry phase: (2000, 3000), wet phase: (4000, 5000)
pContact	probability of contact with another vesicle	(10^−5^, 10^−6^) × (100/currentVolume)^2^
mergeP	probability of absorbing another vesicle	See Table 3 for details
conjugateP	probability of joining peptides or RNAs after absorbing or merging	See Table 4 for details
fitnessScore	jointly determined by the number of AAs assigned (NA) and the total number of peptides (NP) within the vesicle	*FS* = *NA* × *NP*
pSurvival	vesicles’ probability of survival	pSurvival = ln(fitnessScore)/10
rndIndexAA	type of AAs within a vesicle	V1–V5: AA_1_–AA_10_, V6: AA_1_–AA_20_
pABS	probability of absorbing more AAs and NNs from the pond	5 × 10^−3^
pSynchronize	probability of assigning one AA to one set of codons within a cycle	0.999

Following Klein et al. [18], every entity (i.e., amino acid, lipid, polypeptide, nucleic acid, or vesicle) is normalized to a ball-like entity, with a radius of 1 and a mass of 1. Vesicles can absorb amino acids, nucleotides, lipids, and other useful ingredients or components, which are assumed to have an unlimited supply within each pond. The ratio between peptides and RNAs within a vesicle is set to be 2.5–3.5 (e.g., if a vesicle has 2–4 peptides, then it has 5–10 RNAs). Moreover, a vesicle must contain at least 2–3 peptides and 5–10 RNAs in order to qualify as a vesicle (i.e., V1 is the minimum threshold of being a vesicle). Finally, different vesicles have different numbers of AAs being assigned a proper set of codons.

There are six types of vesicles in our simulation (Table 2), and FUCAs are protocells with at least 41–50 peptides and 120–160 RNAs. In other words, only type V6 vesicles are counted as FUCAs. FUCAs, however, have yet to possess a fully evolved SGC [5].

LUCA is a fully functional (proto-)cell with at least one hundred proteins (peptides) and several hundred RNAs (as genes). Moreover, LUCA has a fully evolved SGC, symbolized by the tight coupling of nucleotides (within RNAs) and amino acids (within peptides). In other words, only FUCAs that have evolved the tight linkage of RNA and peptide can become LUCA. Most likely, RNA and proteins co-evolved with each other [43].

Vesicles’ encounters and interactions are driven by wet-and-dry or other similar cycles [7,9,44,45], which is captured by the increased-and-decreased volume of the “warm little pond”. The pond’s changing volume in turn drives changes in the probability of vesicles coming into contact with each other. When in contact, a vesicle can acquire another vesicle according to the probabilities dictated in Table 2.

When two vesicles are merged or one absorbs the other, the new vesicle gains all the peptides and RNAs from its two ancestral vesicles. For simplicity, we assume that peptides and RNAs within the vesicles are fully functional but are agonistic about their specific functions. Moreover, the peptides and RNAs within the new vesicle can be joined to form a longer molecule (i.e., two peptides forming a longer peptide, or two RNAs forming a longer RNA). The probabilities of joining different peptides and RNAs are set in Table 4A and Table 4B, respectively.

Consistent with the thesis that there were two phases of SGC evolution [38,46,47,48,49,50,51,52], we dictate that a vesicle or protocell must first get the 10 early AAs assigned with their respective codons, and only after can a vesicle or protocell get the 10 late AAs assigned with their codons (The ten early AAs are: Gly, Ala, Ser, Asp, Glu, Val, Leu, Ile, Pro, and Thr. The ten late AAs are Phe, Tyr, Arg, His, Trp, Asn, Gln, Lys, Cys, and Met). Consistent with the co-evolutionary theory of codons, proteins, and cellular functions, the more AAs are assigned to their codons, the more functions a protocell will gain from more complex peptides or proteins. Accordingly, a vesicle or protocell should gain a bit of fitness with each “optimal assignment” of AA to its proper codons.

The net fitness score (*FS*) of a vesicle is jointly determined by the number of AAs assigned (*N_A_*) and the total number of peptides (*N_P_*) within the vesicle. That is, *FS = N_A_* × *N_P_*.

For vesicles that do not go through the processes of merger and acquisition during the wet-and-dry cycles, those vesicles with higher fitness scores will have a greater probability of surviving in the pond than those with lower fitness scores. The probability of survival ($P_{S}$) is determined by the simple equation below:$$P_{S}=\frac{\ln\left(FS\right)}{10}$$

Finally, when a FUCA has assigned all the 20 AAs (i.e., $N_{A}=20$) and has more than 100 peptides (i.e., $N_{P}\geq 100$) and 300 RNAs, it becomes a LUCA, and the simulation ends.

### 2.1. The First Stage: The Origin(s) of FUCAs

For the first stage, we start all simulations with only V1 and V2 vesicles.

In the initial state, there were anywhere between 2000 and 3000 V1 and V2 vesicles within a pond of 100 units of volume. For each “cycle” within a simulation, the pond also produces or gains more V1 and V2 vesicles. During the wet phase, anywhere between 4000 and 5000 new V1 and V2 vesicles will be added to the pond, whereas during the dry phase, anywhere between 2000 and 3000 new V1 and V2 vesicles will be added. Vesicles can also absorb other biochemical components and ingredients (e.g., amino acids, nucleotides) and thus become larger vesicles. Biochemical components and ingredients such as amino acids, nucleotides, and lipids are exogenously generated and added to the pond with unlimited supply to speed up the simulations.

Within each cycle, a vesicle has a certain probability of coming in contact with another vesicle. The probability of being in contact is regulated by the total volume of the pond, dictated by the following function: $P_{C}={\left(10\right)}^{-5\sim -6}{\left(\frac{100}{X}\right)}^{2}$, with X being the unit of volume of the whole pond at a given cycle. We further assume that in the dry phase, the total volume of the pond decreases from 100 units of volume to 50–80 units of volume, thus increasing the probability of contact by vesicles. The dry phase also breaks some vesicles. Reversely, in the wet phase, the total volume of the pond increases back to 80–100 units of volume, thus decreasing the rate of contact by vesicles but also allowing new vesicles to form.

When two vesicles come in contact with each other, they can merge with or acquire each other with certain probabilities specified in Table 3. And when smaller vesicles (e.g., V1, V2, and V3) merge with or acquire each other, they form larger vesicles (e.g., V3, V4, and V5).

### 2.2. The Second Stage: From FUCAs to LUCA

It is now widely accepted that LUCAs possessed about several hundred genes, about half of which were in RNA metabolism and translation, with the SGC fully in place. LUCAs also possessed about 140 proteins or domains [36,37,38,48,52,53,54]. We thus assume that LUCA must have about 100–120 proteins and 300–360 RNAs, with each protein being more than 50 AAs and each RNA being more than 150 nucleotide bases long. We also require that LUCA have assigned all twenty AAs to the full genetic code.

For this second stage, we assume that only FUCAs (i.e., V6) can perform the task of assigning the ten late AAs to their proper codons. We thus start with different FUCAs (i.e., V6) that already possess different numbers of peptides and RNAs, and the simulation only selects for LUCA as FUCAs that can eventually produce a complete SGC by assigning codons to AAs, one codon to one AA for each wet-and-dry cycle (or two “cycles” in our simulation). Again, vesicles can merge and acquire other vesicles following the same rules as set in Table 3 and Table 4 in the first stage.

For the second stage, we also assume the following rules:-For simplicity, we skip the evolution of the stop codon. Hence, when all of the 20 AAs have been assigned to their codons, the full SGC has evolved, and we can consider that LUCA now exists. Accordingly, there may be a few LUCAs and they have different genomes, but they all have the same SGC. Very critically, according to our theory, SGC could have only evolved by drawing from “global inventions” via the merger and acquisition of vesicles [5,15,16,17,22] (in this sense, we agree with Herron’s [55] stand that SGC should not be classified as a distinct “major transition” because the evolution of SGC has been a (gradual) process [2]. Notably, while Maynard Smith and Szathmáry [2] identified SGC and protocell (FUCA) as two distinct transitions, Szathmáry later argued that they might have co-evolved in prokaryotic cells. Our thesis holds that they had evolved mostly together, most likely by LUCA [56]).-Taking cues from the thesis that SGC was a “frozen accident” [50,52,57,58,59,60], we assume that there might have been 4 to 6 nucleotides available for making into the SGC before the SGC was finally fixed. Thus, the total combinations of 20 AAs with the possible codons are anywhere between $20(4^{3})$ and $20(6^{3})$, or 1280 to 4320. Thus, for each simulation, the exact number of possible combinations to be assigned is a random number anywhere from 1280 to 4320. For a whole wet-and-dry cycle, FUCAs can only assign one AA to one set of codons, with a fixed probability of 0.999. We are keenly aware that biologically and mathematically, there is a positive feedback mechanism in the evolution of SGC: once one AA has been assigned a set of codons, the remaining AAs will have a smaller pool of codons to be assigned. Hence, an earlier assigning of one AA to a set of codons will accelerate the next cycle of assigning the remaining AAs and codons. As a result, if the probability of assigning the first AA with a set of codons is 10^−3^, the probability of assigning the next AA with the remaining sets of codons becomes: ${P\;(codon-assign)=10}^{-3}{\left(\frac{20}{20-n}\right)}^{2}$, with *n* denoting the number of AAs already assigned. So, for the first AA to be assigned, *n* is 0, and for the second AA, *n* is 1, and so on. The ratio ${\left(\frac{20}{20-n}\right)}^{2}$ is used to magnify the cumulative impact of previous rounds of assigning upon the remaining synchronizations. We initially hoped to implement such dynamics in the simulation. However, due to the fact that many vesicles will die in each cycle (as predicted by our theory [5]), implementing such dynamics requires significant computational resources. We therefore set the probability of successful codon assignment to a fixed probability of 0.999 to speed up the process. Most likely, decreasing the probability will merely prolong the process without fundamentally changing the overall results.-For simplicity, we assume that when two vesicles with different AAs already assigned to their specific codons merge with each other, the merged vesicle obtains all the codons, and hence the evolution of the universal codon accelerates. For example, vesicle-1 has A1, A2, A3, A4, and A5 assigned, whereas vesicle-2 has A1, A2, A3, A4, and A6, then the merged vesicle of the two vesicles will have A1, A2, A3, A4, A5, and A6 assigned. This is consistent with the dynamics underscored by Vetsigian et al. [61] that SGC had most likely evolved via “collective evolution” by drawing from “global innovations” with “horizontal gene transfer” (HGT). Indeed, according to Tang [5], absorption, acquisition, and merger by vesicles entail extensive “horizontal biomolecule transfer” (HBMT) rather than merely HGT: HBMT thus subsuming HGT because HBMT entails exchange and retention of biological ingredients other than genetic materials. Of course, if two vesicles have the same set of AAs assigned (i.e., when both vesicle-1 and vesicle-2 have A1, A2, A3, A4, and A5 assigned), the newly merged vesicle of the two vesicles does not gain a new AA assigned. But the new merged vesicle can still gain peptides and RNAs, thus also increasing its fitness score according to *FS = N_A_* × *N_P_*.

Figure 3 summarizes the evolutionary phase of FUCAs to LUCA.

### 2.3. The Two Stages Together

To test our theory in a complete setting, we also simulate the two stages together. Again, vesicles can merge and absorb each other. Moreover, all vesicles (or protocells) can absorb and integrate amino acids, short peptides, nucleotides, and short RNA molecules into their existing peptides (or proteins) and RNA molecules according to the probabilities dictated in Table 1, Table 2 and Table 3. We also simulate different volumes within different ponds.

The purpose of these simulations is to show that LUCA can indeed emerge from simple vesicles (i.e., V1 and V2) as long as vesicles are allowed to merge and absorb each other in addition to absorbing and integrating amino acids, short peptides, nucleotides, and short RNA molecules into their existing peptides (or proteins) and RNA molecules.

Again, because small vesicles inevitably perish or are merged and absorbed by other larger vesicles, if vesicles can only be generated de novo and in situ with each pond, simulation time will be extremely long within the constraint of computational resources. We therefore also add exogenously generated V1 and V2 vesicles to each pond to speed up the process. More specifically, 2000 to 3000 V1 or V2 are added in the dry phase, while 4000–5000 V1 or V2 are added in the wet phase.

## 3. Results

We now present the simulation results.

### 3.1. The First Stage: The Origin(s) of FUCAs

In various settings, including different volumes of the “warm little pond” and different starting numbers of V1 and V2 vesicles, with about 60–70 cycles, the first FUCA (i.e., V6) will be produced. Eventually, with about 100 to 200 cycles, anywhere from 36 to 284 FUCAs will be produced (Table 5). Numbers of vesicles alive (V1–V6) are the number of vesicles alive within the system when simulations are halted. Snapshots of four specific simulations of the first stage are shown in Appendix A.

### 3.2. The Second Stage: The Origin of LUCA

With different numbers of FUCAs (i.e., V6 vesicles), eventually a LUCA will appear, even though different FUCAs become the LUCA in different simulations in different time frames (Table 6). Hence, as long as each FUCA is allowed to assign codons to amino acids in each cycle, LUCA is an inevitable evolutionary outcome (see also Appendix A for visualizations of two actual synchronization simulations.).

### 3.3. The Two Stages Together

With different numbers of V1 and V2 initially, LUCA can emerge with different number of cycles and with different numbers of other protocells or vesicles within a pond. Table 7 summarizes four such simulations. Again, snapshots of four specific simulations of this set of simulations are shown in Appendix A.

### 3.4. Control Simulations: The Two Stages Together

We have also run a set of control simulations with the two stages together. In these simulations, vesicles cannot merge with each other and hence, acquire biomolecules from each other while all initial conditions, parameters, and mechanisms are identical to the four sets of simulations reported in Table 7. We also run these simulations with more cycles than are needed to produce LUCA in our (positive) simulations.

As shown in Table 8, in these control simulations, no FUCA and hence LUCA ever emerges even after more than 1320 to 1440 cycles, which are more than 200 cycles more than the highest number of cycles (1120) needed to produce LUCA in Table 7. These results show that merger and acquisition by vesicles may well be a key, if not indispensable, mechanism that drives the evolution of LUCA, as Tang [5,22] has argued. Screenshots of four specific control simulations are shown in Appendix A. As shown in these figures, when vesicles cannot merge with and acquire each other, no larger vesicles (i.e., V3 to V6) evolve in the system, and only V1 and V2 vesicles exist even with more cycles lapsed.

## 4. Discussion

According to Tang [5], vesicles’ absorption, acquisition, and fusion via breaking and repacking, proto-endocytosis, proto-endosymbiosis, and other similar processes had been a central and powerful force in the pre-Darwinian evolution before LUCA, long before eukaryogenesis [62,63,64,65], because these processes are processes of variation, selection, and retention.

Absorption, acquisition, and merger are processes of variation because they produce different compartmentalization and hence, different crowding, combination, and coevolution of biomolecules within vesicles. Absorption, acquisition, and merger are also processes of selection and retention because via these processes, some molecules will be retained and integrated within vesicles while some will be excluded from vesicles, and some vesicles will no longer exist. Moreover, absorption, acquisition, and merger entail extensive HBMT rather than merely HGT; HBMT thus subsumes HGT. Indeed, only with HBMT could pre-Darwinian evolution have drawn from “global inventions” and overcome the seemingly insurmountable hurdle of bringing “the overwhelming amount of novelty needed to bring modern cells into existence” [66]. This process of producing LUCA via vesicles’ absorption, acquisition, and merger is far more plausible than the scenario that LUCA has to evolve from a single FUCA de novo and in situ [5].

In this paper, we have shown that the central hypothesis advanced by Tang [5] is indeed viable, at least in computer simulation. If so, we can expect that some of the chemical and biological hypotheses advanced by Tang [5] are also plausible. We therefore hope that our study can stimulate laboratory testing of these hypotheses to show that vesicles’ absorption, acquisition, and merger has indeed been a central force in driving the evolution of LUCA.

Finally, our computer simulation lends more support to the thesis that FUCAs and LUCA had most likely evolved from “fluctuating volcanic hot spring pools” (or Darwin’s “warm little ponds”) on land [6,7,44,67,68,69] rather than from alkaline hydrothermal vents in the ocean [70,71,72]. Most critically, terrestrial hot spring pools allow the wet-and-dry cycles and hence can drive the breakup, repackaging, absorption, merger, and acquisition of vesicles whereas hydrothermal vents do not.

## Figures and Tables

**Figure 1 life-15-00075-f001:**
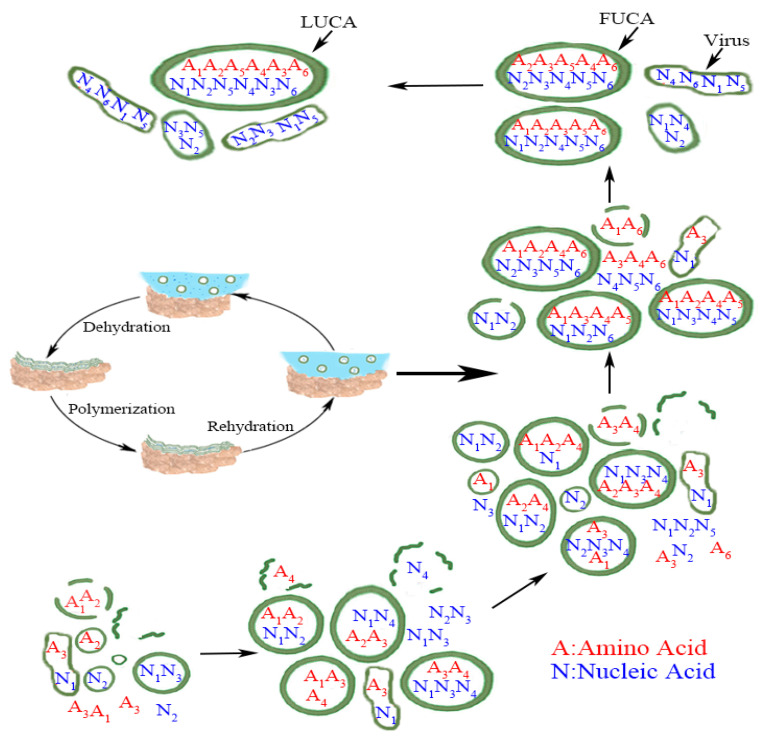
From vesicles to LUCA (reproduced from [5] with permission.) Numbers in subscript denote different amino acids and nucleic acids. The exact matching between amino acid and nucleic acid within LUCA, in a metaphorical sense, implies that the standard genetic code (SGC) had evolved most completely by the time of LUCA. The less than exact matching amino acid and nucleic acid within FUCA and vesicles before FUCA denotes the evolutionary path of SGC from a rudimentary form to a mature form in LUCA. The wet-and-dry cycle part within the figure is adapted from [7] with their kind permission. Figure 1 retains virus to make the whole picture complete, but we do not simulate the evolution of virus here. Credit: author.

**Figure 2 life-15-00075-f002:**
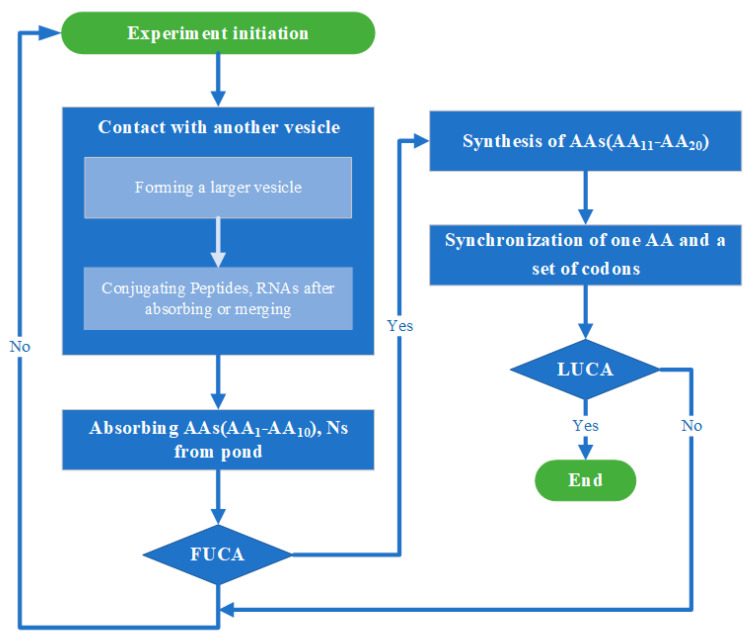
Flowchart of simulation. Credit: author.

**Figure 3 life-15-00075-f003:**
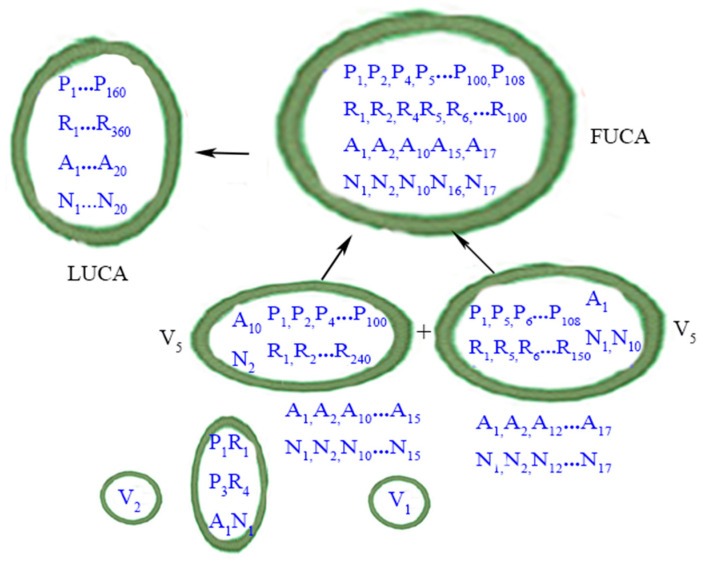
From FUCAs to LUCA. P: peptide; R: RNAs. A: amino acid; N: nucleotide (as codons). Protocells or vesicles are in closed circles. LUCA is depicted according to the definition in the main text. The perfect matching of amino acids (A) and nucleic acids (N) within LUCA, in a metaphorical sense, implies that the standard genetic code (SGC) had evolved most completely by the time of LUCA. The less than exact matching of amino acids and nucleic acids within the one FUCA and the two V5 vesicles denotes the evolutionary path of SGC from a rudimentary form to a mature form in LUCA. Contents within the two smaller vesicles (i.e., V1 and V2) are now shown. The wet-and-dry cycle part is omitted here. Credit: author.

**Table 2 life-15-00075-t002:** Types of vesicles (V1 to V6).

Types of Vesicles	No. of Peptides (N_P_)	No. of RNAs (N_R_)	No. of AAs Assigned (N_A_)	Fitness Score (F)
V1	2–4	5–10	5–6	10–24
V2	5–10	10–30	6–8	30–80
V3	11–20	25–70	8–10	88–200
V4	21–30	50–100	10–12	210–360
V5	31–40	75–140	12–14	272–560
V6 (as FUCAs)	41–50	120–160	14–16	576–800

**Table 3 life-15-00075-t003:** Probabilities of absorbing another vesicle vs. being absorbed (the first number for a vesicle in rows; the second for a vesicle in columns).

	V1	V2	V3	V4	V5	V6
V1	0.5; 0.5	0; 1	0; 1	0; 1	0; 1	0; 1
V2	1; 0	0.5; 0.5	0.4; 0.6	0.3; 0.7	0.2; 0.8	0.1; 0.9
V3	1; 0	0.6; 0.4	0.5; 0.5	0.4; 0.6	0.3; 0.7	0.2; 0.8
V4	1; 0	0.7; 0.3	0.6; 0.4	0.5; 0.5	0.4; 0.6	0.3; 0.7
V5	1; 0	0.8; 0.2	0.7; 0.3	0.6; 0.4	0.5; 0.5	0.4; 0.6
V6	1; 0	0.9; 0.1	0.8; 0.2	0.7; 0.3	0.6 0.4	0.5; 0.5

Note: When two identical vesicles encounter each other, they have an equal probability of acquiring as merging with each other (hence, 0.5 vs. 0.5). The actual probability of an acquisition is drawn randomly around the two numbers within a margin of ±5~10%.

**Table 4 life-15-00075-t004:** (**A**). Rules for joining peptides after absorbing or merging. (**B**) Rules for joining RNAs after absorbing or merging.

(**A**)				
**Length of Peptide (AAs)**	**3–10**	**11–25**	**26–50**	**>51**
3–10	1 × 10^−5^	0.6 × 10^−5^	0.3 × 10^−5^	0.1 × 10^−5^
11–25	0.6 × 10^−5^	0.3 × 10^−5^	0.1 × 10^−5^	0.05 × 10^−5^
26–50	0.3 × 10^−5^	0.1 × 10^−5^	0.05 × 10^−5^	0.02 × 10^−5^
>51	0.1 × 10^−5^	0.05 × 10^−5^	0.02 × 10^−5^	0.01 × 10^−5^
(**B**)				
**Length of RNA (NBs)**	**6–30**	**31–60**	**61–100**	**>100**
6–30	1 × 10^−5^	0.6 × 10^−5^	0.3 × 10^−5^	0.1 × 10^−5^
31–60	0.6 × 10^−5^	0.3 × 10^−5^	0.1 × 10^−5^	0.05 × 10^−5^
61–120	0.3 × 10^−5^	0.1 × 10^−5^	0.05 × 10^−5^	0.02 × 10^−5^
>120	0.1 × 10^−5^	0.05 × 10^−5^	0.02 × 10^−5^	0.01 × 10^−5^

**Table 5 life-15-00075-t005:** The first stage: evolution of FUCAs under different settings.

Indicators\Simulations	1	2	3	4	5	6	7	8
Volume of the pond (in units)	60	72	66	69	52	93	85	62
No. of V1 vesicles alive	1863	1845	1888	1928	1784	1608	1624	1568
No. of V2 vesicles alive	1867	1849	1895	1891	1785	1610	1621	1552
No. of V3 vesicles alive	926	926	944	980	906	903	897	880
No. of V4 vesicles alive	265	280	288	317	374	439	438	471
No. of V5 vesicles alive	71	77	89	98	120	171	174	239
Total No. of vesicles (V1–V6) alive	5028	5019	5149	5266	5039	4865	4893	4994
No. of V6 (i.e., FUCAs) produced	36	42	45	52	70	134	139	284
Cycles needed for the first FUCA to evolve	112	115	118	125	128	140	142	170

Note: in all simulations with GAMA, 1 “cycle” = 1 wet phase or 1 dry phase, alternated.

**Table 6 life-15-00075-t006:** From FUCAs to LUCA (Stage 2): starting with 300 FUCAs.

Simulation	FUCAs		No. of Cycles for Producing LUCA	No. of LUCA Produced	LUCA	
	No. of Peptides	No. of RNAs			No. of Peptides	No. of RNAs
1	(100, 389)	(300, 1127)	50	1	239	851
2	(100, 416)	(300, 1204)	56	8	(212, 416)	(679, 1204)
3	(101, 290)	(300, 834)	54	1	253	768
4	(100, 276)	(300, 843)	54	1	276	627
5	(100, 297)	(300, 886)	54	1	230	675
6	(100, 277)	(300, 831)	56	4	(246, 277)	(672, 823)
7	(100, 404)	(300, 1222)	54	1	286	719
8	(100, 294)	(302, 805)	56	2	(253, 294)	(655, 756)
9	(100, 276)	(300, 827)	56	2	(249, 276)	(751, 827)
10	(100, 389)	(300, 1198)	54	1	215	721
11	(100, 280)	(300, 859)	56	1	249	778
12	(100, 280)	(300, 836)	56	2	(210, 261)	(693, 780)
13	(100, 297)	(300, 801)	54	1	289	736
14	(100, 290)	(300, 846)	52	1	223	836
15	(100, 292)	(300, 875)	54	1	217	716
16	(100, 296)	(300, 842)	54	1	231	694
17	(100, 294)	(300, 849)	54	1	244	715
18	(100, 275)	(300, 880)	56	1	259	807
19	(100, 392)	(300, 1275)	56	1	274	787
20	(100, 266)	(300, 832)	54	1	264	728
21	(100, 263)	(300, 806)	56	1	227	723
22	(100, 293)	(300, 867)	54	1	232	793
23	(100, 403)	(300, 1149)	56	2	(264, 403)	(858, 1149)
24	(100, 401)	(300, 1193)	56	4	(248, 272)	(714, 874)
25	(100, 347)	(300, 1246)	54	1	251	683

Note: FUCAs are generated randomly with different numbers of peptides and RNAS, within a range. Different simulations produce LUCA with different number of peptides and RNAs, at different cycles. The key result is that eventually a LUCA will appear, even though different FUCAs evolve into LUCAs in different simulations.

**Table 7 life-15-00075-t007:** Summary of four simulations with the two stages together.

Parameters\Simulations	1	2	3	4
Volume of the pond (in units)	58	76	55	57
No. of V1s and V2s alive when simulations are halted	1169; 2036	1014; 1812	985; 1839	993; 1885
Percentage of peptides or proteins longer than 50 AAs in LUCA	80%	80%	80%	80%
Percentage of RNAs longer than 150 NBs in LUCA	70%	75%	78%	80%
Total No. of protocells perished	2,946,377	3,712,377	3,083,724	4,085,174
Total No. of alive protocells when the first LUCA emerged	3510	3125	3212	3337
No. of cycles until the first LUCA emerged	800	1000	840	1120
No. of FUCA produced	50	71	59	67
No. of LUCA produced	1	1	1	1

**Table 8 life-15-00075-t008:** Summary of four control simulations with the two stages pooled together.

Parameters\Simulations	1	2	3	4
Volume of the pond (in units)	53	88	83	80
No. of V1s and V2s alive when simulations are halted	1163; 1952	1476; 2187	1588; 2459	1663; 2327
Percentage of peptides or proteins longer than 50 AAs in a LUCA	N.A.	N.A.	N.A.	N.A.
Percentage of RNAs longer than 150 NBs in a LUCA	N.A.	N.A.	N.A.	N.A.
Total No. of protocells perished	4,646,260	4,590,189	5,024,335	4,836,936
Total No. of alive protocells when simulation is halted	3115	3663	4047	3960
No. of cycles when simulation is halted	1330	1320	1440	1360
No. of FUCA produced	0	0	0	0
No. of LUCA produced	0	0	0	0

Note: Because no LUCA will ever emerge from these control simulations, we cannot know the length of peptide or RNA within a possible LUCA. Hence, row 4 and row 5 are not really applicable (N.A.) here. They are retained for comparison only.

## Data Availability

The original contributions presented in the study are included in the article/Appendix A, further inquiries can be directed to the corresponding author. All computer codes and experimental data are available online. https://github.com/xymbgm/2023-05-Tang-and-Gao-Simulation-of-from-FUCA-to-LUCA-codes (accessed on 1 October 2024).

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
