# Peer review of "The Origin(s) of LUCA: Computer Simulation of a New Theoryâ€"

_life, 2025, doi:10.3390/life15010075_

Round 1
Reviewer 1 Report
Comments and Suggestions for Authors
The work is high-quality and very important for advancing the understanding of the emergence of biological systems; however, it is based on two self-citations, while the coiners of the pivotal term have been left out as one of the many references included in one of the mentioned self-citations. May I suggest that the discussion of the current work could be enhanced even more by including that earlier contentious paper? Perhaps by reading it more carefully to hold the thrust of that earlier proposal, finding so the commonalities and disagreements between the two works (from the coiners and from the current important 'in silco' work).
Also, I tried to run the script in the GAMA platform but I found some difficulties that I thing that could be overcome by clarifying better the metholodogy, becasue so far I could not reproduce your results and I hope it is due to some missing instructions.
Additionally, I found a lack of controls (positive and/or negartive).
Lastly, there are a few small typos.
You can find all the observations in the attached document.

Author Response
We have replied to R1's comments in the document, Reply to R1 comments-life-3270335

Reviewer 2 Report
Comments and Suggestions for Authors
Simulation results and data in this manuscript are suffiecient enough for readers in order to repeat or modify the simulation. I have few comments , respected Author can find the below
1. Table 1: Please remove "See Table 2" and add the range of the numbers. Information in each table should be provided completely and independently from other tables.
2. Are all 20 types of amino acids have been used uniformly in each simulation? and how about nucleic acids? Any simulations results with less than 20 amino acids involved?
3. If temperature and pressure plays role in the simulstion formulas, please state them in the text.
4. Refering to the results of Table 6, 20 out of 300 theoretical FUCAs had a chance to evolved as LUCAs. Please state that then the probability (better to use the concept of frequency) is 1/15, which is significant.
Author Response
We deeply appreciate R2 for her/his positive assessment of our contribution.
To make the replication of our simulations more straightforward, for this version, we have labelled all the simulation codes with their corresponding tables or figures in the github.
For detailed replies, see the attached file, Reply to R2 comments-life-3270335

Reviewer 3 Report
Comments and Suggestions for Authors
Points related to biological justification:
1. The paper presents a model in which vesicles contain different sets of peptides and RNAs and partially complete genetic codes. These gradually combine by processes of cell merger until a cell is produced containing the complete standard genetic code. This cell is called the LUCA. I find it fairly plausible that cells could build up in this way and that the complexity of cells could increase due to mergers and horizontal transfer of molecules between cells. However, it is not clear what is actually being tested by the model in this paper. The model seems to be designed in such a way that this outcome is bound to happen. What have we learned? In particular, it is claimed that cell mergers and horizontal transfer are essential to lead to the creation of the LUCA. But there seems to be no control simulation without these features. I would have liked to see a case where there is just vesicle division and vertical transfer (no mergers) in order to see whether the LUCA can arise in this case. I would expect that it can still arise in this case, but it might be true that it takes much longer than if mergers and horizontal transfers occur.
2. Figure 1 includes Viruses. What exactly is a virus in this model? In the figure, it seems to be a vesicle with nucleic acids and no proteins. Can this virus multiply? It does not seem to be a parasite of the functional cells, as is the case for modern viruses. Is there any distinction in the model between a virus, and a vesicle with a small incomplete set of peptides and RNAs? It may be better to avoid the word virus altogether.
Points related to model design:
1. Please spell out more clearly what distinguishes FUCA from LUCA, and what distinguishes a FUCA from a smaller vesicle. Lines 61-65 state properties that LUCA is assumed to have. The model seems to say that only type V6 vesicles are FUCAs, but why are these different from V1-V5? In Figure 2 it seems to be that only a FUCA (V6) can assign a codon for one of the later 10 amino acids, but why is this? What actually happens when a codon is assigned in the genetic code? There must be a tRNA and an aminoacyl-tRNA synthetase that is newly created. But it is not stated whether the RNAs and peptides in the cell actually have these functions. The later amino acids are those which were not abundant in prebiotic chemistry. Therefore, a cell which uses these later amino acids would have to have an enzymatic process that synthesizes them. Are the peptides in the cell supposed to have these enzymatic functions?
2. Why do there need to be six distinct categories of vesicles 1-6? Can you not simply define the survival and merger probabilities vesicles as smooth functions of the numbers of molecules in the vesicles without needing to divide them into categories?
3. In Table 3, I have not understood the difference between absorbing another vesicle and being absorbed. If they combine to a single vesicle, the result does not depend on which one absorbs the other.
4. If I understand correctly, Np and Nr in table 2 are the numbers of peptides and RNAs in a vesicle and you are assuming these are specific sequences of amino acids or nucleotides (i.e. they have a function, they are not just random sequences). Are these the number of distinct molecules with different functions, or are they the total number of copies? How do you make new copies of these functional molecules? I am imagining a cell in which RNA replication and translation are occurring, so you get multiple copies of each molecule. But I don’t understand whether the number of copies of each molecule in a cell is tracked by the program.
5. I don’t see any vesicle division in the model description. Surely division must occur sometimes to increase cell number, to balance the decrease in dumber when cells merge? When division occurs, it is necessary to know how many copies of each molecule are contained in the parent cell. If there is only one copy of each molecule, the daughter cells will obviously have fewer distinct molecules than the parent.
6. Line 237 says that the pond produces more V1 and V2 vesicles. Where do these vesicles come from? And where do the peptides and nucleic acids in these vesicles come from? Would it not be better to create new vesicles by division rather than just by popping up out of nowhere?
7. I am not sure what is meant by ‘conjugation’. I think you are talking about joining two sequences to make a longer one (but I am not sure). Why do these probabilities in Tables 4A and 4B depend on the lengths of the sequences? I also wondered whether conjugation referred to transfer of a molecule from one cell to the other – as in bacterial conjugation, which can lead to transfer of a plasmid from one cell to another.This would be relevant for a model of protocells, but it does not seem to be what is meant here (???)
Points related to presentation:
1. The graphs in figures 4, 5, 6 are not legible. It is not possible to see which curve is which. The small writing in the side bars is not legible, and is probably not necessary. Please replot these with a graph drawing program, rather than showing screen shots from the simulation package.
2. Fig 5 says the y axis is the number of sets of codons to be assigned. But aren’t there only 64 codons to start with? Why is the y axis more than 64?
3. Type setting on line 246 is not clear. What is the mathematical function for Pc?
Author Response
We thank R3 for his/her careful reading and constructive comments. In this version, we have tried to improve as much as possible by taking R3’s comments into consideration.
We have replied to R3's comments in detail in the attached file, Reply to R3 comments-life-3270335

Round 2
Reviewer 1 Report
Comments and Suggestions for Authors
I am glad to see the improvement in the quality of research with the incorporation of controls, and with the instructions clearly stated in the 'readme' document.
I can appreciate the supplementary material with all the graphics, keeping the main text only with tables and the models depicted.
As I mentioned in the first evaluation, this work is highly relevant in the field as this is the first, to my knowledge, to prove how FUCA could have appeared and therefore the LUCA emergence from FUCA states. I agree with authors in that FUCA was not an isolated entity but comprised lipids, amino acids, and other fundamental components.
I do not have any additional comments, there is no need of an attached document.
I highly recommend the publication of this work as it is of paramount importance in the field of early emergence of pre-/proto-biological systems.
Author Response
We thank R1 for her/his wonderful support for the revised version. We are also glad that R1 has found the supplememtary material and appendix to be clear and useful.
Since R1 demands no further revising, we have no further replies to R1.
Reviewer 3 Report
Comments and Suggestions for Authors
I will not oppose publication of the modified version. But there are still many questions that are not clear to me and I do not find the paper very convincing.
I will just insist on one important point that I am still unclear about. I think there is some misunderstanding between me and the authors about what is meant by division. The authors insist that they do not want to put in division. By division I simply mean cell reproduction. Surely there must be new cells being formed and these must acquire molecules from previous cells. If the new cells are accurately replicated, they will have the same contents as the parent before division. I cannot see how you can have any kind of evolutionary model without reproduction/division. If this is not occurring, how can any new properties that arises in one cell ever spread through the population. The addition of the control simulations shows that there is no emergence of LUCA without cell mergers – but if there is still no division in the control experiments, I just don’t get the point. You can't expect a LUCA to emerge in a single randomly created cell without evolution at all.
A new point is that I am also confused about lines 295-297. For four bases and codons of length 3, we have 4^3 =64 codons. Then 20 x 64 = 1280. But if each codon could be assigned to any amino acid, the number of possible genetic codes is 64^20 = 1.3 x 10^36, not 1280. It says ‘the number of codons to be assigned is a random number anywhere from 1280 to 4320’. But the number of codons to be assigned is 64, not 1280. I don’t understand where 1280 comes in.
Author Response
Please see the attachment, "Reply to R3 comments on the revised version-08-12-2024"
